# Impact of Patient-Reported Outcome Monitoring via Recovery Tracker on Post-Discharge Outcomes After Colorectal Surgery: A Comparative Analysis Before and After Implementation [note 1]

**DOI:** 10.3390/cancers17121939

**Published:** 2025-06-11

**Authors:** Hans M. Huber, Iris H. Wei, Mohammad Ali Abbass, Georgios Karagkounis, Maria Widmar, J. Joshua Smith, Garrett M. Nash, Martin R. Weiser, Philip B. Paty, Julio Garcia-Aguilar, Emmanouil Pappou

**Affiliations:** Department of Surgery, Memorial Sloan Kettering Cancer Center, New York, NY 10065, USA; hans.huber@mssm.edu (H.M.H.); abbassm1@mskcc.org (M.A.A.); widmarm@mskcc.org (M.W.);

**Keywords:** patient-reported outcomes, remote symptom monitoring, Recovery Tracker, readmission, urgent care visits, digital health

## Abstract

When patients go home after colorectal surgery, they may experience symptoms that can lead to anxiety or unplanned hospital visits. To help monitor recovery, our hospital introduced an electronic system called the Recovery Tracker, which allows patients to report symptoms daily from home. We compared patients treated before and after the system was introduced to see if it helped reduce unnecessary hospital visits. We found that while overall visit rates increased, patients who used the Recovery Tracker were less likely to be readmitted to the hospital. This suggests that patients who stay engaged with symptom reporting may have better recoveries. Our findings highlight the importance of supporting patients in using these tools and point to future possibilities, like combining symptom reports with data from wearable devices. This research can help guide how hospitals use technology to keep patients safer and more supported during their recovery at home.

## 1. Introduction

Patients recovering at home after surgery face a critical gap in the monitoring and support of their symptoms, which can be a considerable source of stress [1,2]. Pain, worrisome symptoms, or complications may arise during this period, yet patients often lack the clinical expertise to discern normal recovery from warning signs of serious issues, and the resulting uncertainty can cause substantial distress [3,4,5,6]. Furthermore, inadequately-managed postoperative symptoms can lead to unnecessary emergency room visits and interruptions in cancer-related care such as initiation of chemotherapy [2,7].

In recent years, patient portals and electronic patient-reported outcome (PRO) monitoring tools have allowed patients to report symptoms in real time and communicate with providers remotely. Previously published literature has described the usage of technology platforms for enhanced communication between providers and patients in the management of long-term diseases [8,9]. Multiple trials have demonstrated the acceptability of electronic symptom-reporting systems among patients undergoing treatment for malignancy [10,11,12].

Electronic patient portals have been utilized to provide postoperative follow-up visits and gather data on at-home recovery with a high degree of patient acceptability [10,13]. In the field of colorectal surgery, several small studies have reported an 80–90% response rate to remote electronic symptom monitoring questions [7,14]. These findings suggest that leveraging PRO-based smartphone applications is a scalable strategy to monitor recovery, detect complications sooner, and potentially prevent avoidable readmissions. Within the past two decades, the adoption of Enhanced Recovery After Surgery (ERAS) after colorectal surgery has resulted in shorter inpatient hospitalizations and a larger percentage of the recovery period spent at home [15,16,17,18].

To address the post-discharge monitoring gap, our institution developed the Recovery Tracker (RT) in 2016 as a web-based electronic survey system for surgical patients. The RT delivers a series of daily symptom questionnaires (for 10 days after surgery) via the patient portal, and provides immediate feedback indicating whether reported symptoms are expected or if follow-up is required. Initial implementation of the RT focused on ambulatory cancer surgeries at a specialty center, and early results were encouraging. A prior study reported that use of the RT in ambulatory surgery patients was associated with fewer unplanned urgent care visits [19].

Building on that success, in March 2021 the RT platform was expanded to include patients undergoing inpatient colorectal surgery—a population with longer operative times and higher complexity than the ambulatory cohort. The use of the RT for this patient population represented a new role for the RT platform. It was hypothesized that integrating PRO-driven daily symptom tracking in this higher-risk group would facilitate early identification of issues and thereby reduce unplanned hospital visits after discharge. In the present study, we compare post-discharge outcomes before and after RT implementation in patients undergoing elective inpatient colorectal surgery, with a particular focus on how patient-reported symptom survey engagement relates to urgent care visits and readmissions.

## 2. Materials and Methods

The Recovery Tracker system is a series of 10 question surveys distributed daily via the MyMSK Patient Portal. Patients are notified about surveys via email, and are sent a daily survey for the first 10 days after discharge. A full listing of the survey questions is presented in Table 1. The system is programmed to alert the surgery team if prespecified severity levels are met. For intermediate symptoms such as worsening pain, a notification of a “yellow alert” goes to the care team. For critical symptoms such as intractable vomiting or severe shortness of breath, a “red alert” is sent. Yellow alerts trigger a notification to the surgery office team and follow up via phone during normal business hours. Red alerts prompt the patient to immediately contact the surgery office and concurrently notifies the on-call surgery team at all hours, including nights and weekends. Phone follow-up is handled by nursing staff during normal business hours and the on-call colorectal fellow during nights, weekends, and holidays.

This was a retrospective observational study conducted at a single, high-volume cancer care center. Clinical research staff and medical doctors abstracted data from the institutional database via chart review. All data points were initially collected as part of routine clinical care. The experimental “Recovery Tracker” cohort consisted of all patients undergoing elective, inpatient colorectal surgery between 1 March 2021 (the date of RT implementation) and 31 December 2022. Inclusion criteria were as follows: (1) underwent elective operation performed by a surgeon within the colorectal department; (2) were admitted as an inpatient for at least 1 night; (3) were enrolled in the MyMSK Patient Portal system in order to receive RT surveys; and (4) were 18 years or older at the time of surgery.

A control cohort of patients who had undergone elective inpatient colorectal surgery between 1 February 2019 and 1 March 2020 was identified from the institutional database by the same clinical research staff. Inclusion criteria for this group were the same as the experimental cohort. The dates of the control cohort were chosen to avoid potential confounding effects due to the onset of the COVID-19 pandemic in March 2020. Exclusion criteria for both the experimental and control cohort included admissions for urgent/emergent surgery and age under 18 at the time of surgery.

Demographic characteristics were tested for equivalence between the two cohorts using chi-squared test for categorical variables and unpaired *t*-test for continuous variables. A *p*-value < 0.05 was deemed statistically significant for the purpose of this study. See Table 2 for a complete listing of demographic characteristics and associated *p*-values.

The primary aim of this study was to assess whether there were significant differences in the rate of readmission and urgent care center (UCC) visits without admission in the 30 days following discharge between the RT and control cohorts. Readmission was defined as any admission to our hospital within the first 30 days following discharge after elective colorectal surgery. A UCC visit was defined as any visit to the urgent care center not resulting in an inpatient admission within 30 days of discharge. Readmissions more than 30 days after discharge were deemed to be beyond the scope of this study and were not included. A multivariable logistic regression analysis was performed using backwards elimination to test the associations of RT with the primary outcomes while adjusting for demographic variables. The goal of this analysis was to test whether the odds of a UCC visit or readmission differed between colorectal patients before and after RT implementation.

We also hypothesized that there would be a difference in readmission rates between patients in the RT cohort who utilized and did not utilize the RT platform. An additional sub-group analysis was performed among the groups to assess if higher engagement with the RT was associated with lower odds of readmission or UCC visit. Using multivariable regression with backwards elimination, we tested for an association between survey response rate and readmission/UCC visit within 30 days of discharge. Statistical significance was set at <0.05 for all regression tests in this study. All statistical analyses were performed using SAS v 9.4 (SAS Institute, Cary, NC, USA).

## 3. Results

### 3.1. Patient Characteristics

A total of 1941 patients who were enrolled in the Recovery Tracker (RT) met the inclusion criteria. All patients in this cohort had undergone elective colorectal surgery requiring inpatient admission between March 2021 and December 2022. The median age in this group was 59 (IQR 50–69) and 899 (46.3%) were female. The median BMI of the RT cohort was 26.4 (IQR 23.0–30.4). A total of 1548 (79.8%) of the patients were designated as ASA 3 or higher, 888 (45.7%) patients had open operations, and 463 (23.9%) patients had an ostomy at the time of discharge. Median operative length was 201 min (IQR 18–282).

A total of 1206 patients who underwent elective colorectal surgery between February 2019 and March 2020 were included in the control group. The median age of this cohort was 60 (IQR 50–70) and 566 (46.9%) were female. Median body mass index (BMI) was 26.9 (IQR 23.2–30.9) and 967 (80.2%) had an ASA score of 3 or higher. Overall, 542 (44.9%) patients had an open operation as opposed to minimally invasive or robotic surgery, and 261 (21.6%) were discharged with an ostomy in place. The median OR time for the control cohort was 204 min (IQR 134–307). Demographic, clinical, and operative variables for both cohorts are further described in Table 2. Within the RT cohort, there was a statistically significant higher rate of discharge with an ostomy in place (23.9% vs. 21.6%, *p* = 0.0019). No other demographic variables demonstrated statistically significant discrepancies between cohorts.

CPT codes were used to determine the frequency of each operation in the overall cohort. The three most commonly performed operations were right hemicolectomy (23.4%), low anterior resection/sigmoidectomy (16.3%), and left hemicolectomy (15.9%). There were lower proportions of right hemicolectomy (22.1% vs. 25.5%, *p* = 0.0338), LAR/sigmoidectomy (15.1% vs. 18.2%, *p* = 0.0254), and abdominoperineal resection (4.4% vs. 6.4%, *p* = 0.0159) in the RT cohort compared to the control cohort. Readmission rates by procedure are presented in Table 3 and Table 4 for both the RT cohort and the control cohort.

### 3.2. Response Rates and Alerts

Among the RT cohort patients, 1362 (70.2%) responded to at least one of the RT surveys that were assigned. The median number of surveys completed was three with an interquartile range of 0–7.579 (29.8%) patients responded to no surveys and were designated as “non-responders” for the purposes of this study. A total of 996 (51.3%) patients responded to three or more surveys. A total of 7374 surveys were completed out of 18,553 assigned (a completion rate of 39.7%). Overall, 5772 (78.3%) surveys triggered no alerts based on responses, 1373 (18.6%) triggered a “yellow” alert, and 229 (3.1%) triggered a “red” alert. Of the entire recovery tracker cohort, 156 (7.0%) unique patients triggered a red alert and 513 (26.6%) triggered a yellow alert.

### 3.3. Primary Outcome Analysis

In the RT cohort, 86 (4.43%) presented to the UCC but were not readmitted to the hospital within 30 days of discharge. Within the control cohort, 20 (1.61%) patients had a UCC visit without readmission. On multivariable analysis, having surgery during the post RT implementation period was associated with an odds ratio of 2.795 (95% CI 1.707–4.577, *p* < 0.0001) of having a UCC visit. Two covariates were also significantly associated with increased risk of UCC visits: having an ASA score of 3 or more (OR 2.407, 95% CI 1.244–4.656, *p* = 0.0091) and operating room time in minutes (OR 1.002, 95% CI 1.000–1.0004, *p* = 0.0103). Further details of the multivariable analysis are presented in Table 5.

Having surgery in the RT period was also associated with a higher risk of overall 30 readmissions to the hospital. The rate of readmission in the post RT group was 9.74% compared to 6.88% in the control group (OR 1.431, 95% CI 1.091–1.877, *p* = 0.0098). Three other covariates were associated with readmission on multivariable analysis. These included an ASA score of 3 or more (OR 1.600, 95% CI 1.083–2.362, *p* = 0.0181), having an open as opposed to a minimally invasive operation (OR 1.850, 95% CI 1.425–2.403, *p* < 0.0001), and having an ostomy in place at the time of discharge (OR 1.852, 95% CI 1.417–2.419, *p* < 0.0001). Female sex was associated with lower odds of readmission (OR 0.749, 95% CI 0.580–0.969, *p* = 0.0275).

### 3.4. Subgroup Analysis

A multivariable analysis was also performed comparing patients in the RT cohort who responded to at least one survey (responders) vs. those who completed no surveys (non-responders). Among the 1362 patients who completed at least one survey, 56 (4.11%) had a UCC visit without readmission, compared to 30 (5.18%) of the 579 patients who completed no surveys. However, the odds of UCC visit did not reach statistical significance on multivariable analysis of the RT cohort (OR 0.868, 95% CI 0.546–1.381, *p* = 0.5505). an ASA score of 3 or more and operation length were significant covariates affecting UCC visit (Table 6).

Overall, 30-day readmission rates demonstrated a statistically significant difference between these two subgroups. A total of 85 (6.24%) responders were readmitted compared to 56 (9.67%) non-responders (OR 0.561, 95% CI 0.410–0.767, *p* = 0.0003). Other significant covariates for readmission in this subgroup analysis included female sex, ASA score above 3, and presence of an ostomy at discharge (Table 6). Patients who responded to three or more surveys had a readmission rate of 5.2% compared patients who completed two or fewer surveys. This result was statistically significant in a multivariable regression model (OR 0.378, 95% CI 0.268–0.531, *p* < 0.0001).

## 4. Discussion

In this cohort study, engaging with the PRO-based Recovery Tracker was associated with improved outcomes in postoperative colorectal surgery care. Patients who responded to at least one RT survey had 44% lower odds of 30-day readmission compared to those who never responded, identifying non-responders as a high-risk subgroup. This effect was amplified with higher response rates: patients who engaged more frequently with the RT by filling out three or more surveys had 62% lower odds of readmission compared to all others. The RT cohort experienced a higher rate of urgent care center (UCC) visits without admission (4.43%) compared to the control cohort (1.61%). Additionally, the 30-day readmission rate was higher in the RT cohort (9.74%) versus the controls (6.88%). Significant risk factors for readmission included female sex, ASA score above 3, and presence of an ostomy at discharge.

Non-responders were identified as a high-risk group in this study, with significantly higher rates of 30-day readmissions compared to responders. This suggests that patients who do not engage with the Recovery Tracker (RT) system may require additional support to optimize their postoperative outcomes. Non-response could indicate barriers such as limited technological proficiency, lack of awareness about the importance of symptom monitoring, or greater underlying vulnerability to complications. Tailored interventions, such as proactive nurse-led check-ins, simplified digital tools, or targeted education efforts may help address these challenges and improve engagement among non-responders. Identifying and intervening early in this subgroup could enhance recovery trajectories and reduce the burden of hospital readmissions, emphasizing the need for a more personalized approach to postoperative care.

Despite the intuitive appeal of remote symptom monitoring, our data did not show an overall reduction in post-discharge healthcare utilization with the RT system in this cohort. Unexpectedly, the patients in the RT implementation period had higher rates of urgent care center (UCC) visits and 30-day readmissions compared to a historical control group. This outcome contrasts with our initial hypothesis and with earlier observations in the ambulatory setting, where introduction of the Recovery Tracker was associated with fewer non-emergent hospital visits after surgery. It is noteworthy that a randomized trial of electronic symptom monitoring after ambulatory cancer surgery also found no significant differences in avoidable UCC visits or readmissions between patients who received an ePRO intervention and those who did not. Collectively, these findings suggest that while electronic PRO systems can improve certain aspects of care, their impact on hard clinical endpoints like emergency visits and readmissions may depend on multiple factors. The higher utilization seen in our RT cohort could reflect the greater complexity of inpatient colorectal surgery patients, who inherently have more postoperative issues than ambulatory surgical patients. It is also possible that the RT facilitated earlier detection of problems, resulting in patients being appropriately routed to urgent evaluation (thus increasing UCC visits) in situations that previously might have gone unnoticed until a later, perhaps more serious readmission. In other words, an increase in acute care visits is not necessarily negative if it means patients are seeking care at the right time and level of need.

Our findings align with the previous literature demonstrating the benefits of electronic patient-reported outcomes (ePROs) in enhancing communication and symptom management during recovery [20,21,22,23,24,25]. Prior studies have shown that the RT system reduces unnecessary urgent care visits in ambulatory surgery patients, including those undergoing breast, urologic, and gynecologic procedures [19,26]. The higher rate of UCC visits among the RT cohort in our data may reflect the differences between the inpatient colorectal population and those undergoing outpatient procedures. While our data did not clearly point to a single factor associated with the UCC visit increase, it is possible that usage or PRO systems has a different effect on recovery for patients who are discharged after longer hospital stays and who undergo more complex operations. The lower readmission rates among RT responders in our study mirror findings from prior research demonstrating the positive impact of ePRO systems on reducing hospital readmissions. [19,22]. Studies have highlighted that real-time monitoring enables early identification of complications, facilitating timely intervention and reducing symptom severity [23]. However, our findings also emphasize that the effectiveness of RT may vary across surgical populations, with inpatient colorectal surgery patients presenting unique challenges compared to ambulatory cohorts.

Even if the impact on utilization is equivocal, electronic symptom monitoring appears to confer important benefits for patient experience and clinical management. Participants in a qualitative evaluation of the RT described the tool as an “extension of care”, noting that it improved communication with their providers and prompted reflection on their recovery [27]. The automated feedback was especially valued for providing reassurance and setting appropriate expectations about what symptoms were “normal”. This aligns with findings from the ACCESS trial, in which the arm of patients who received real-time normative feedback on their symptoms reported significantly lower postoperative anxiety and required fewer nurse phone calls for symptom management [27,28]. Collectively, these insights highlight that PRO platforms like the Recovery Tracker can help patients feel more supported and less anxious during the vulnerable post-discharge period. Patients know that their daily reports are being reviewed and that concerning symptoms will trigger a response, effectively creating a safety net. One interviewed patient likened the RT to having a “virtual hand on your shoulder”, underscoring the psychological reassurance it provides. On the other hand, these systems are not without limitations. When a patient experienced a symptom that was not listed in the RT questionnaire, it could cause confusion or stress about what to do next. This points to the importance of continuously refining electronic PRO tools to cover a comprehensive range of common postoperative concerns and to educate patients on how to report or seek help for unexpected issues. It also reinforces the need for human backup: if a patient’s survey responses stop or indicate severe trouble, the care team should actively follow up. In our study, the ease of use of the RT was generally high, but ensuring equitable access and usability for all patient demographics is paramount. Notably, older patients and those with less tech experience might require additional training or alternative reporting options (such as telephone-based PRO reporting) to engage fully with such programs.

Future research should focus on developing tailored interventions to enhance engagement among non-responders and optimizing the alert system to reduce unnecessary healthcare utilization. Proactive strategies such as automated follow-ups for non-responders, integrating additional educational resources, and providing culturally sensitive and accessible interfaces could improve adoption and outcomes. Additionally, wearable devices and passive tracking technologies represent a promising future direction in postoperative monitoring. By integrating biometric data such as step count, heart rate variability, sleep patterns, and activity levels, wearables can provide real-time physiologic context to patient-reported outcomes. These tools could help identify deviations from expected recovery trajectories even in patients who are less likely to complete active symptom surveys. Combining PRO data with continuous passive monitoring may enhance clinical responsiveness, personalize recovery goals, and further reduce preventable readmissions. As mobile health technologies evolve, future implementations may feature seamless integration of ePRO platforms with wearable data streams to create comprehensive, adaptive recovery dashboards for both patients and providers.

It is important to note that our findings may not be fully generalizable to other healthcare settings due to the robust nurse follow-up protocols already in place at our institution. In our system, nurses routinely call patients post-discharge to monitor recovery, provide guidance, and address concerns, ensuring a high level of patient oversight. This baseline practice likely mitigates some of the benefits that might otherwise be observed with the use of the Recovery Tracker system. Consequently, the RT system may not have demonstrated a substantial difference in outcomes or reductions in unnecessary urgent care visits in our study. In healthcare systems without such proactive nurse-led follow-up, the RT system may play a more critical role in supporting postoperative recovery and reducing healthcare utilization. These factors should be considered when interpreting the findings and assessing the applicability of the RT system in other practice environments.

There are several other limitations to our study that warrant consideration. First, the retrospective design introduces potential biases and limits causal inference. Second, our analysis relied on data from within our institutional network, and urgent care visits or readmissions occurring outside this system were not captured. Third, all patients enrolled in the RT cohort were required to use the MyMSK patient portal, which may introduce selection bias favoring patients with higher technological literacy and socioeconomic status. Finally, the study was conducted in a single, high-volume cancer center, limiting the generalizability of our findings to other healthcare settings.

## 5. Conclusions

In summary, integrating patient-reported outcome monitoring into post-discharge surgical care holds significant promise, but its success depends on patient engagement and thoughtful implementation. The Recovery Tracker system exemplifies how a PRO platform can extend postoperative surveillance beyond the hospital, potentially improving patient satisfaction and safety at home [29]. While our retrospective analysis did not find a drop in readmissions or urgent visits after RT implementation, it did identify a clear benefit for those patients who actively used the tool, and it revealed a subset of non-responders who may benefit from additional support. These results echo the mixed findings in the literature—some studies show reductions in complications or healthcare use with ePRO monitoring, whereas others show no change in hard outcomes but improvements in patient-reported metrics like anxiety and symptom control. Going forward, maximizing the impact of such systems will likely require a combination of technological and workflow refinements, for example, integrating PRO alerts with triage pathways, adding educational modules for common postoperative issues, and instituting protocols to reach out to patients who do not report. Nonetheless, even in their current form, electronic PRO tools provide a valuable adjunct to surgical follow-up. They empower patients to participate in their own recovery, provide clinicians with an early warning of complications, and foster a sense of connectedness in the critical days after discharge. As healthcare moves toward value-based models, widespread adoption of PRO-based remote monitoring could improve outcomes in a cost-effective manner. The experience from our Recovery Tracker program will inform future efforts to refine these systems, with the ultimate goal of safer recoveries and better patient-centered outcomes after surgery.

## Figures and Tables

**Table 1 cancers-17-01939-t001:** Questions assigned to colorectal surgery patients via Recovery Tracker (RT) survey.

Symptom	Question Prompt	Response Type	Title 1	Title 2	Title 3	Title 4
Shivering	How often did you have shivering or shaking chills?	Likert Scale	entry 1	data	data	data
Fever	Have you had a temperature greater than 100.4 °F or 38 °C?	Yes/No		data	data	data
Pain	How would you describe your pain?	Likert Scale		data	data	data
Nausea	How would you describe your nausea (feeling like you were going to throw up)?	Likert Scale	entry 2	data	data	data
Vomiting	How much did you vomit (throwing up) in the past 24 h?	Likert Scale		data	data	data
Appetite	How much have you been eating?	Likert Scale	entry 3	data	data	data
Drinking less	How much have you been drinking?	Likert Scale		data	data	data
Constipation	When was your last bowel movement (poop)?	Time (1–3+ days)	entry 4	data	data	data
Shortness of Breath	How would you describe your shortness of breath (trouble breathing)?	Likert Scale		data	data	data
Redness	Have you noticed redness around your surgical site(s) getting more red or bigger?	Yes/No				
Discharge	How severe has the discharge been (fluid leaking from your wound) at the surgical incision(s)?	Likert Scale				
Ambulation	Compared to how you were before this surgery, how limited are you today in walking inside your home?	Likert Scale				
Fatigue	How would you describe your fatigue?	Likert Scale				

**Table 2 cancers-17-01939-t002:** Patient characteristics.

Characteristic	Pre-RT (n = 1206)	Post-RT (n = 1941)	*p* Value
Age, median, years (IQR)	60 (50–70)	59 (50–69)	0.2209
Female (%)	566 (46.9)	899 (46.3)	0.7409
BMI, median, kg/m^2^ (IQR)	26.9 (23.2–30.9)	26.4 (23.0–30.4)	0.1082
ASA score (%)			
1–2	239 (19.8)	393 (20.2)	
3+	967 (80.2)	1548 (79.8)	0.6808
Open operation (%)	542 (44.9)	888 (45.7)	0.6855
OR time, median, minutes (IQR)	204 (134–307)	201 (128–282)	0.1904
Discharged with ostomy (%)	261 (21.6)	463 (23.9)	0.0019
Length of stay, median, days (IQR)	4 (3–7)	4 (3–8)	0.1454
**Type of operation (%)**			
Right Hemicolectomy	307 (25.5)	429 (22.1)	0.0338
Left Hemicolectomy	186 (15.4)	313 (16.1)	0.6160
LAR/Sigmoidectomy	219 (18.2)	293 (15.1)	0.254
Abdominoperineal Resection	77 (6.4)	85 (4.4)	0.0159
Ostomy Reversal	111 (9.2)	221 (11.4)	0.0561
Other	306 (25.4)	600 (30.9)	0.0005

**Table 3 cancers-17-01939-t003:** RT cohort readmissions by procedure.

Operation	*N *(% of Entire Cohort)	Readmissions (% by Operation)	UCC Visits (% by Operation)
Left Hemicolectomy	313 (16.1)	20 (6.4)	17 (5.4)
Right hemicolectomy	429 (22.1)	24 (5.6)	13 (3.0)
Low anterior resection	317 (16.3)	29 (9.1)	15 (4.7)
Total Abdominal Colectomy	26 (1.3)	4 (15.4)	0 (0.0)
Abdominoperineal resection (APR)	87 (4.5)	18 (20.7)	5 (5.7)
Ostomy reversal	228 (11.7)	12 (5.2)	10 (4.4)

**Table 4 cancers-17-01939-t004:** Control cohort readmissions by procedure.

Operation	*N* (% of Entire Cohort)	Readmissions (% by Operation)	UCC Visits (% by Operation)
Left hemicolectomy	189 (15.3)	10 (5.3)	0 (0.0)
Right hemicolectomy	313 (25.3)	21 (6.7)	5 (1.6)
Low anterior resection	230 (18.6)	15 (6.5)	5 (2.2)
Total abdominal colectomy	21 (1.7)	3 (14.2)	1 (4.8)
Abdominoperineal resection (APR)	81 (6.5)	11 (6.5)	1 (1.2)
Ostomy reversal	111 (9.0)	7 (6.3)	1 (0.9)

**Table 5 cancers-17-01939-t005:** Multivariable logistic regression model predicting odds of UCC visit without admission and overall 30-day readmission.

Characteristic	UCC Visit Without Readmission	30-Day Readmission	Data	
	OR (95% CI)	*p* Value	OR (95% CI)	*p* Value
Tracker enrollment				
Control cohort	Reference	NA	Reference	NA
RT cohort	**2.795 (1.707–4.577)**	**<0.0001**	**1.431 (1.091–1.877)**	**0.0098**
Age	1.005 (0.990–1.019)	0.5148	0.996 (0.987–1.006)	0.4251
Sex				
Male	Reference	NA	Reference	NA
Female	1.138 (0.767–1.688)	0.5219	**0.749 (0.580–0.969)**	**0.0275**
BMI	1.013 (0.979–1.048)	0.4600	0.999 (0.987–1.006)	0.9536
ASA score				
1–2	Reference	NA	Reference	NA
3+	**2.407 (1.244–4.656)**	**0.0091**	**1.600 (1.083–2.362)**	**0.0181**
Open operation	1.204 (0.803–1.806)	0.3685	**1.850 (1.425–2.403)**	**<** **0** **.0001**
Operation length (minutes)	**1.002 (1.000–1.003)**	**0.0103**	1.001 (1.000–1.002)	0.0904
Discharged with ostomy	1.403 (0.914–2.152)	0.1211	**1.852 (1.417–2.419)**	**<** **0** **.0001**

Statistically significant covariates highlighted in **bold.**

**Table 6 cancers-17-01939-t006:** Multivariable logistic regression model tracker cohort with responder analysis.

Characteristic	UCC Visit Without Readmission	30-Day Readmission	Data	
	OR (95% CI)	*p* Value	OR (95% CI)	*p* Value
Tracker Response				
Non-responders	Reference	NA	Reference	NA
Responders	0.868 (0.546–1.381)	0.5505	**0.561 (0.410–0.767)**	**0.0003**
Age	1.004 (0.988–1.020)	0.6518	1.010 (0.998–1.021)	0.1012
Sex				
Male	Reference	NA	Reference	NA
Female	1.086 (0.699–1.686)	0.7141	**0.551 (0.400–0.759)**	**0.0003**
BMI	1.008 (0.971–1.047)	0.6666	1.003 (0.976–1.030)	0.8444
ASA score				
1–2	Reference	NA	Reference	NA
3+	**2.869 (1.311–6.277)**	**0.0083**	1.645 (1.002–2.700)	0.0489
Open operation	1.222 (0.773–1.933)	0.3911	**2.120 (1.541–2.917)**	**<** **0** **.0001**
Operation length (minutes)	**0.999 (0.997–1.000)**	**0.0304**	0.999 (0.998–1.000)	0.2675
Discharged with ostomy	0.998 (0.606–1.644)	0.9952	**2.052 (1.467–2.833)**	**<** **0** **.0001**

Statistically significant covariates highlighted in **bold.**

## Data Availability

Deidentified data are present on the institutional server.

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
