# Peer review of "Impact of Patient-Reported Outcome Monitoring via Recovery Tracker on Post-Discharge Outcomes After Colorectal Surgery: A Comparative Analysis Before and After Implementationâ€"

_cancers, 2025, doi:10.3390/cancers17121939_

Round 1
Reviewer 1 Report
Comments and Suggestions for Authors
I read with interest the manuscript by Huber et al. titled “Impact of Patient-Reported Outcome Monitoring via Recovery Tracker on Post-Discharge Outcomes After Colorectal Surgery: A Comparative Analysis Before and After Implementation.” The manuscript addresses the relevant topic of outcome monitoring, offering meaningful insights into postoperative and post-discharge outcomes following colorectal surgery.
Please see my comments reported in a point-by-point manner:
- The introduction is overly long. I recommend reducing its length by removing specific details about the Recovery Tracker and the treating center, which would be more appropriately placed in the methods section.
- The language should be reviewed to ensure scientific precision. For instance, the sentence “Patients who had undergone non-elective or emergent operations (such as surgery for perforated colon cancer) were excluded” could be more clearly stated as “Exclusion criteria included urgent or emergency surgery.” I suggest a full revision of the inclusion and exclusion criteria to ensure clarity and scientific rigor.
- The methods section is underdeveloped and disproportionately short compared to the introduction. A detailed and transparent methodology is essential for reproducibility. Key elements are missing, such as the definitions of outcomes (e.g., readmission within how many days?), and the specification of primary and secondary endpoints.
- In the results section, the cohorts should be described and compared at baseline. Variables such as age, BMI, and comorbidities should be reported with appropriate statistical comparisons (e.g., p-values). It is not scientifically valid to compare outcomes between two cohorts without first demonstrating whether they are comparable at baseline.
Author Response
- The introduction is overly long. I recommend reducing its length by removing specific details about the Recovery Tracker and the treating center, which would be more appropriately placed in the methods section.
In this draft we have pared down the introduction to eliminate redundant language. We also moved descriptive sentences regarding the recovery tracker to the methods section.
- The language should be reviewed to ensure scientific precision. For instance, the sentence “Patients who had undergone non-elective or emergent operations (such as surgery for perforated colon cancer) were excluded” could be more clearly stated as “Exclusion criteria included urgent or emergency surgery.” I suggest a full revision of the inclusion and exclusion criteria to ensure clarity and scientific rigor.
The authors reviewed the entirety of the manuscript to eliminate redundant or unclear language and ensure specificity. We added a sentence clearly describing exclusion criteria in a concise manner.
- The methods section is underdeveloped and disproportionately short compared to the introduction. A detailed and transparent methodology is essential for reproducibility. Key elements are missing, such as the definitions of outcomes (e.g., readmission within how many days?), and the specification of primary and secondary endpoints.
The authors edited the methods section to ensure clarity about the primary outcome and the subgroup analysis looking at readmission rates based on whether or not a patient utilized the recovery tracker. Timeline for the outcome of interest (within 30 days of discharge after elective colorectal surgery) was also described.
- In the results section, the cohorts should be described and compared at baseline. Variables such as age, BMI, and comorbidities should be reported with appropriate statistical comparisons (e.g., p-values). It is not scientifically valid to compare outcomes between two cohorts without first demonstrating whether they are comparable at baseline.
We added the P values from our statistical testing (chi square and unpaired T tests) for differences in demographic variables to an expanded table 1. In the results section we describe discrepancies between the two cohorts to ensure transparency.
Reviewer 2 Report
Comments and Suggestions for Authors
Dear Authors,
This retrospective study provides valuable insights into the impact of electronic patient-reported outcomes (ePRO) on urgent care utilization and readmission rates following elective colorectal surgery. The study design is clearly articulated, and the results are well presented.
However, the discussion section does not address the observed increase in urgent care center visits and 30-day readmission rates within the remote triage (RT) cohort. Including an explanation or possible interpretation of these findings would strengthen the discussion.
Overall, this is a well-executed and presented study, but the references should be reviewed again.
Author Response
"However, the discussion section does not address the observed increase in urgent care center visits and 30-day readmission rates within the remote triage (RT) cohort. Including an explanation or possible interpretation of these findings would strengthen the discussion."
This question is now addressed in paragraph four of the discussion. We focus on the fact that inpatient colorectal surgery represents a very different cohort of patients compared to outpatient procedures. Therefore, the impact of PRO notification systems may differ based on what type of surgery/hospital course a patient had.
"Overall, this is a well-executed and presented study, but the references should be reviewed again."
References were reviewed by the authors and moved in several instances to ensure relevance.
Reviewer 3 Report
Comments and Suggestions for Authors
This trial examines the usefulness of monitoring with a recovery tracker to evaluate the condition of patients after discharge from the hospital after undergoing colorectal cancer surgery. This paper is considered to be a very good attempt.
Major comments
1: The evaluation is based on service provided during normal business hours only, but is there no service provided at night or on holidays? If the service had been available at night and on holidays, its usefulness may have been even greater. Please state whether there were any alerts from the Recovery tracker during the night or on holidays.
2: Although a responder is defined as someone who only answers one question in the Recovery tracker, is there any difference based on the number of questions answered?
Author Response
"The evaluation is based on service provided during normal business hours only, but is there no service provided at night or on holidays? If the service had been available at night and on holidays, its usefulness may have been even greater. Please state whether there were any alerts from the Recovery tracker during the night or on holidays."
"Although a responder is defined as someone who only answers one question in the Recovery tracker, is there any difference based on the number of questions answered?"
We performed a further analysis looking at readmission rates among patients who answered three or more surveys using multivariable logistic regression. These patients had a lower odds of readmission compared when compared with our analysis of those who answered only one survey. This finding is now discussed in the results section.